# Grape Seed Proanthocyanidins Induce Apoptosis and Cell Cycle Arrest of HepG2 Cells Accompanied by Induction of the MAPK Pathway and NAG-1

**DOI:** 10.3390/antiox9121200

**Published:** 2020-11-28

**Authors:** Lihua Wang, Jicheng Zhan, Weidong Huang

**Affiliations:** Beijing Key Laboratory of Viticulture and Enology, College of Food Science and Nutritional Engineering, China Agricultural University, Beijing 100083, China; lihuawang@cau.edu.cn

**Keywords:** hepatocellular carcinoma, grape seed proanthocyanidins, apoptosis, cell cycle, mitogen-activated protein kinases, non-steroidal anti-inflammatory drug-activated gene-1

## Abstract

Hepatocellular carcinoma (HCC) is one of the common malignancies leading to death. Although radiotherapy and chemotherapy have certain effects, their side effects limit their therapeutic effect. Phytochemicals have recently been given more attention as promising resources for cancer chemoprevention or chemotherapy due to their safety. In this study, the effects of grape seed proanthocyanidins (GSPs) on the apoptosis, cell cycle, and mitogen-activated protein kinase (MAPK) pathway-related proteins and non-steroidal anti-inflammatory drug-activated gene-1 (NAG-1) expression of HepG2 cells were investigated. The results showed that GSPs inhibited the viability of HepG2 cells in a time- and dose-dependent manner, induced apoptosis and G2/M phase cell cycle arrest, and regulated cell cycle-related proteins, cyclin B1, cyclin-dependent kinase 1, and p21. GSPs also increased reactive oxygen species production and caspase-3 activity. In addition, GSPs also increased the expression of p-ERK, p-JNK, p-p38 MAPK and NAG-1, and GSPs-induced NAG-1 expression was related to the MAPK pathway-related proteins. These data suggest that GSPs may be promising phytochemicals for HCC chemoprevention or chemotherapy.

## 1. Introduction

Hepatocellular carcinoma (HCC) is one of the common malignancies as a major threat to human health worldwide and accounts for 90% of primary liver cancers [1]. Effective prevention and treatment measures are currently unavailable [2]. HCC is usually treated with surgery, such as liver resection or transplantation, and medical treatments, such as chemoradiation and sorafenib [3,4,5]. Surgical resection is still the most effective treatment for early stage liver cancer, but most patients with malignant tissues do not have characteristic clinical manifestations. When their tumors are discovered, they are already in an advanced stage, and many patients miss the best opportunity for surgical resection [5]. Although radiotherapy and chemotherapy have certain effects, their side effects or drug resistance limit their therapeutic effect [6].

Available studies have shown that proanthocyanidins exhibit antitumor effects on a variety of human cancers [7], such as head and neck squamous cancer [8], non-small cell lung cancer [9,10], colorectal cancer [11,12], HCC [13,14] and pancreatic cancer [15]. Studies have been demonstrated that proanthocyanidins are safe within a certain dose range in vitro and in vivo [16]. Many foods (various fruits, beans, chocolates, etc.) and beverages (fruit juice, wine, beer, tea, etc.) are rich in proanthocyanidins, and the most abundant source is grape seeds [17]. Grape seed proanthocyanidins (GSPs) are a mixture of dimers, trimers, tetramers and oligomers/polymers by the polymerization of catechins and/or epicatechins [18].

Many studies have shown that drug-induced reactive oxygen species (ROS) production plays a vital role in the apoptosis of different types of cancer [19]. Like many drugs that induce cancer cell apoptosis through ROS, proanthocyanidins can also induce apoptosis in cancer cells through ROS. Hop proanthocyanidins trigger apoptosis in human colorectal adenocarcinoma cells [20], grape seed extract induces apoptosis in Caco-2 human colon cancer cells [19], and procyanidins from *Vitis vinifera* seeds induce apoptotic cell death in squamous cell carcinoma cells [21] through ROS.

The MAPKs family has three major members, namely, ERK, JNK and p38 MAPK [22]. It has been found that the MAPK pathway-related proteins play an essential role in cancer growth, apoptosis and metastasis [23]. NAG-1, a transforming growth factor-β superfamily member, has proapoptotic and antitumorigenic activity [24]. NAG-1 transgenic mice show anticancer effect in the colon and lung [25,26]. It has been reported that MAPK pathway-related proteins are involved in the regulation of NAG-1 expression [27,28,29]. Studies have shown that NAG-1 plays an important role in regulating the apoptosis of cancer cells [29,30,31]. In addition, studies have also shown that NAG-1 is involved in the cell cycle regulation [32] and cell migration of cancer cells [33].

Although studies have reported that proanthocyanidins can trigger the apoptosis of cancer cells [14,21,34,35], their effects on the expression and modulation of mitogen-activated protein kinases (MAPK) pathway-related proteins and non-steroidal anti-inflammatory drug-activated gene-1 (NAG-1), as well as their relationship in HepG2 cells, still remain unclear. In this study, the effects of GSPs on proliferation, apoptosis, cell cycle, the MAPK pathway-related proteins, and NAG-1 expression of HepG2 cells were investigated. This study provides supporting evidence for GSPs as promising phytochemicals for HCC chemoprevention or chemotherapy.

## 2. Materials and Methods

### 2.1. Reagents and Antibodies

Dulbecco’s Modified Eagle Medium (DMEM, 4.5 g/L D-glucose, L-glutamine and 110 mg/L sodium pyruvate), RPMI-1640 medium, fetal bovine serum, penicillin streptomycin (10,000 units/mL penicillin and 10,000 μg/mL streptomycin), and 0.25% trypsin-EDTA were obtained from Gibco, Thermo Fisher Scientific, Inc. (Waltham, MA, USA). GSPs (purity ≥ 95.0%) were obtained from Chengdu Must Bio-technology Co., Ltd. (Chengdu, Sichuan, China) and dissolved in dimethyl sulfoxide (DMSO). SB203580, PD98059 and SP600125 were obtained from Beyotime Biotechnology (Shanghai, China).

Antibodies against extracellular regulated protein kinase (ERK), c-Jun N-terminal kinase (JNK), p38 MAPK, NAG-1, cyclin-dependent kinase 1 (CDK1), cyclinB1, p21 and GAPDH were obtained from Proteintech Group, Inc. (Wuhan, Hubei, China). Antibodies against p-ERK (Thr202/Tyr204), p-JNK (Thr183/Tyr185) and p-p38 MAPK (Thr180/Tyr182) were obtained from Cell Signaling Technology (Boston, MA, USA). 

### 2.2. Cell Culture

The HCC cell lines (HepG2 and SMMC-7721) were generously gifted from Prof. Hongbo Hu (China Agricultural University, Beijing, China). HepG2 cells and SMMC-7721 cells were cultured in DMEM and RPMI-1640 medium supplemented with 10% heat-inactivated fetal bovine serum and 1% penicillin streptomycin, respectively, and in a humidified incubator containing 5% CO_2_ at 37 °C.

### 2.3. MTT Assay for Cell Viability and Colony Formation Assay 

The MTT assay was used to measure the viability of HepG2 and SMMC-7721 cells. The cells were plated into 96-well plates at a density of 5000 cells/well in 100 μL medium supplemented with 10% heat-inactivated fetal bovine serum and 1% penicillin streptomycin. After 24 h of incubation in a humidified incubator containing 5% CO_2_ at 37 °C, HepG2 cells were treated with GSPs (5, 10, 20, 30 and 40 mg/L), and SMMC-7721 cells were treated with GSPs (20, 40, 60, 80 and 100 mg/L) or equivalent amounts of DMSO (0.1%) as the negative control. After 24 or 48 h of treatment, MTT (Shanghai Sangon Biotech Co., Ltd., Shanghai, China) solution (5 mg/mL) was added to each well, followed by incubation for 4 h. The supernatants were then removed and the formazan precipitates were dissolved in 150 μL of DMSO. The absorbance was measured at 490 nm with a microplate reader (Thermo Fisher Scientific, MA, USA). Cell viability was calculated according to the following formula: cell viability (%) = (treatment group OD − blank group OD)/(negative control OD − blank group OD) × 100%.(1)

For the colony formation assay, HepG2 cells were seeded into 6-well culture plates (300 cells/well), cultured for 24 h, and treated with or without 10 mg/L of GSPs for 24 h. After 14 days of culture, the colonies were fixed in methanol and stained with 0.005% crystal violet (BBI Life Sciences, Shanghai, China).

### 2.4. Apoptosis Analysis Using Flow Cytometry

The effects of GSPs on apoptosis in HepG2 cells was analyzed using flow cytometry according to the instructions of TransDetect Annexin V-FITC/PI Cell Apoptosis Detection Kit (Transgen Biotech, Beijing, China). Briefly, cells were seeded in 6-well plates and cultured for 24 h in a humidified incubator containing 5% CO_2_ at 37 °C. After cells were treated with 10 mg/L of GSPs for 24 or 48 h, adherent cells collected by trypsinization and floating cells were collected by centrifugation. These collected cells were washed twice with cold PBS and resuspended using 100 μL of ice-cold 1 × Annexin V Binding buffer followed by a mix of 5 μL of Annexin V-FITC and 5 μL of PI. The cells were then incubated in the dark at room temperature (20–25 °C) for 15 min before 400 μL of ice-cold 1 × Annexin V Binding buffer was added. Finally, the stained cells were detected using a FACSCalibur flow cytometer (BD biosciences, San Jose, CA, USA), and the data were analyzed by the Cell Quest Software (BD Biosciences).

### 2.5. Determination of ROS Production

ROS production in HepG2 cells was measured according to the protocol of the ROS assay kit (Beyotime Biotechnology, Shanghai, China). Briefly, fluorescent probe DCFH-DA was diluted with serum-free medium to a final concentration of 10 μM. The cells treated with or without 10 mg/L of GSPs for 24 h were collected and suspended in diluted DCFH-DA, and incubated in a 37 °C cell incubator for 20 min. Then, the cells were washed three times with serum-free medium to fully remove the DCFH-DA that had not entered the cells. Finally, the fluorescence of the cells was measured using a FACSCalibur flow cytometer (BD biosciences, San Jose, CA, USA), and the data were analyzed by the Cell Quest Software (BD Biosciences).

### 2.6. Measurement of the Mitochondrial Membrane Potential (MMP)

Changes in the MMP of the cells treated with or without 10 mg/L of GSPs for 24 h were measured using flow cytometry according to the protocol of the MMP assay kit with JC-1 (Beyotime Biotechnology, Shanghai, China). Briefly, cells were collected and resuspended in cell culture medium. The cells were then incubated with the fluorescent lipophilic cationic probe JC-1 at 37 °C for 20 min. Subsequently, the cells were centrifuged, washed twice with JC-1 staining buffer, and resuspended with an appropriate amount of JC-1 staining buffer. Finally, the fluorescence of the cells was measured using a FACSCalibur flow cytometer (BD biosciences, San Jose, CA, USA), and the data were analyzed by the Cell Quest Software (BD Biosciences). 

### 2.7. Determination of Caspase-3 and Caspase-9 Activity

Cells treated with or without 10 mg/L of GSPs for 24 h were lysed in cold lysis buffer and centrifuged at 14,000× *g* for 10 min at 4 °C. Caspase-3 and caspase-9 activity was determined using a commercial caspase-3 and caspase-9 activity assay kit (Beyotime Biotechnology, Shanghai, China). Briefly, 20 μL of cell lysate was mixed with 40 μL of caspase buffer and 10 μL of Ac-DEVD-pNA (2 mM) or Ac-LEHD-pNA (2 mM) and incubated at 37 °C for 4 h. The absorbance of the samples was then measured at a wavelength of 405 nm. The protein concentration of the samples was determined by the Bradford assay. The caspase-3 and caspase-9 activity contained in the unit weight protein was calculated based on total sample protein concentration. 

### 2.8. Cell Cycle Analysis Using Flow Cytometry

For cell cycle distribution, after treatment with 10 mg/L of GSPs for 24 or 48 h, the floating and adherent cells were collected, washed with cold PBS, and fixed with 70% ethanol at 4 °C for 24 h. The cells were then washed with cold PBS and incubated with 500 μL PI solutions in the dark at 37 °C for 30 min. For each experiment, 2 × 10^4^ cells were recorded. Cell Quest software (BD Biosciences) was used to analyze cell cycle.

### 2.9. RNA Isolation and Quantitative Real-Time Polymerase Chain Reaction (qPCR) Analysis 

HepG2 cells were cultured in 6-well plates, and the total RNA of cells treated with or without 10 mg/L of GSPs was isolated using Trizol (Takara, Dalian, China). The total RNA was evaluated for integrity by agarose gel electrophoresis and for purity by spectrophotometry. cDNA was obtained using a HiFiScript cDNA Synthesis Kit (Cwbio, Beijing, China) according to the manufacturer’s instructions. The GAPDH was used as the internal reference gene. The primer sequences are NAG-1-sense (5′-CAGTCGGACCAACTGCTGGCA-3′), NAG-1-antisense (5′-TGAGCACCATGGGATTGTAGC-3′); GAPDH-sense (5′-TCTGGTAAAGTGGATATTGTTG-3′), GAPDH-antisense (5′-GATGGTGATGGGATTTCC-3′) [29].

Two-step qPCR was carried out using a CFX96 Connect™ Real-Time PCR System (Bio-Rad, Hercules, CA, USA). Each reaction was carried out in triplicate and the reaction volume was 10 μL, containing 3.5 μL of diluted cDNA, 0.2 μL of each primer (10 μM), 5 μL of UltraSYBR mixture (Cwbio, Beijing, China), and 1.1 μL of ddH_2_O. The amplification protocol was predenaturation at 95 °C for 10 min, followed by 40 cycles of amplification including denaturation at 95 °C for 15 s and annealing at 60 °C for 1 min. The melting curve analysis was carried out from 65 to 95 °C. The 2^−ΔΔCT^ method was used to calculate relative gene expression [36]. The reference sample was untreated cells, and it was defined as expression = 1. Relative expression of gene was expressed as the fold-change compared with the reference sample.

### 2.10. Western Blot Analysis

Briefly, cells were washed twice with cold PBS and scraped off using lysis buffer (20 mM Tris (pH 7.5), 150 mM NaCl, 1% Triton X-100, sodium pyrophosphate, β-glycerophosphate, EDTA, Na_3_VO_4_ and leupeptin) (Beyotime Biotechnology, Shanghai, China) containing protease inhibitor PMSF (1 mM). The protein concentration of samples was then measured using a BCA protein assay (Pierce^®^ BCA Protein Assay Kit, Thermo Fisher Scientific, MA, USA). Equal amounts of denatured proteins (20–40 µg/well) were loaded to SDS-PAGE gel electrophoresis and transferred onto polyvinylidene fluoride (PVDF) membranes (Immobilon^®^-P Transfer Membrane, Millipore, Billerica, MA, USA) using wet transfer. The membranes were then blocked with 5% skim milk in TBST at room temperature for 1 h followed by incubation with specific primary antibodies at 4 °C overnight. Each membrane was then washed five times with TBST and incubated with the secondary antibody-horseradish peroxidase (HRP) conjugated anti-rabbit IgG antibody diluted in 5% skim milk at room temperature for 1 h. Each membrane was then washed five times with TBST. Finally, the membranes were exposed to enhanced chemiluminescence reagents to detect the HRP signal. 

### 2.11. Plasmid and Transfection

The CDS region of human NAG-1 was amplified using PrimeSTAR^®^ HS DNA Polymerase (Takara, Dalian, China) from the human cDNA. DNA extraction after PCR was performed using TaKaRa MiniBEST Agarose Gel DNA Extraction (Takara, Dalian, China). The CDS region of human NAG-1 was cloned into a pcDNA3.1 vector digested with the *Kpn*I and *XBa*I restriction enzymes using the ClonExpress II One Step Cloning Kit (Vazyme, Nanjing, China). The clones were sequenced to ensure the accuracy of the PCR amplification of the CDS region of human NAG-1. 

For transfection, in brief, cells were plated onto 6-well plates and grown for 24 h. The cells were co-transfected with 2.5 µg of plasmid constructs using Lipo8000™ Transfection Reagent (Beyotime Biotechnology, Shanghai, China) for 24 h. After transfection for 24 h, cells were lysed to detect the transfection efficiency by western blotting or treated with GSPs for 24 h.

### 2.12. Statistical Analysis

The data are expressed as the mean ± standard deviation (SD) of three independent experiments. Statistical analysis was performed by one-way analysis of variance using SPSS version 21 (SPSS Inc, Chicago, IL, USA). Duncan’s multiple range test was conducted to indicate the significant difference at *p* < 0.05 or *p* < 0.01.

## 3. Results

### 3.1. GSPs Inhibit Proliferation of HepG2 Cells

To investigate whether GSPs could exhibit an antiproliferative effect on HCC, two different cell lines, HepG2 and SMMC-7721 cells, were used for the study using an MTT assay. HepG2 and SMMC-7721 cells were treated with different concentrations of GSPs for 24 or 48 h. As demonstrated in Figure 1A,B, GSPs reduced the viability of HepG2 and SMMC-7721 cells in a dose- and time-dependent manner. Since the treatment of GSPs showed more significant inhibition on HepG2 cells, the HepG2 cells were selected as the cell line for further study. When HepG2 cells were treated with 10 mg/L of GSPs for 48 h, cell viability was reduced by approximately 50%, so 10 mg/L of GSPs was selected as the treatment concentration for further study. Additionally, 10 mg/L of GSPs significantly decreased the number of HepG2 cells colonies (Figure 1C,D), further confirming its antiproliferative activity.

### 3.2. GSPs Induce the Apoptosis of HepG2 Cells

Cellular apoptosis is an important cause of cell death and inhibition of proliferation [37]. Therefore, flow cytometry was used for Annexin V-FITC/PI double staining to analyze apoptosis induced by GSPs. Apoptosis can be divided into early stage apoptosis illustrated in the lower right quadrants of the FACS histograms and late stage apoptosis shown in the upper right quadrants of the FACS histograms. The results showed that 10 mg/L of GSPs treatment for 24 and 48 h induced significant apoptosis (early stage and late stage) of HepG2 cells (Figure 2A,B). When HepG2 cells were treated with 10 mg/L of GSPs for 24 h, there were a greater number of cells in early stage apoptosis than late stage apoptosis (Figure 2A). However, when HepG2 cells were treated with 10 mg/L of GSPs for 48 h, there were a greater number of cells in late stage apoptosis than early stage apoptosis (Figure 2B).

### 3.3. GSPs Trigger ROS Production, Decrease the MMP, and Increase the Caspase-3 Activity of HepG2 Cells

Studies have shown that ROS controls cell growth, proliferation, migration and apoptosis as a “second messenger” in the intracellular signaling cascade [38]. Thus, the effect of GSPs on ROS production was investigated using a DCFH-DA probe. The results showed that 10 mg/L of GSPs dramatically triggered the ROS production of cells treated for 24 h (Figure 3A).

Decreased MMP is a critical event in the early stage of apoptosis [39]. In addition, the main source of ROS in the cells is mitochondria [38]. The change of JC-1 from red fluorescence to green fluorescence can indicate a decrease in membrane potential [40]. Thus, JC-1 was used to determine the MMP of cells treated or untreated with GSPs for 24 h. As demonstrated in Figure 3B, GSPs markedly decreased the MMP of cells treated with 10 mg/L of GSPs. 

Due to decreased MMP, apoptotic-inducing factors, such as cytochrome c and activated caspases, are released into the cytosol [41]. Thus, caspase-3 and caspase-9 activity was measured, and the results suggested that treatment with 10 mg/L of GSPs for 24 h caused a significant increase in caspase-3 activity but had no significant effect on caspase-9 activity of HepG2 cells (Figure 3C).

### 3.4. GSPs Induce G2/M Phase Cell Cycle Arrest of HepG2 Cells

The cell cycle plays an important role in cell growth [42]. To investigate whether the change of the proliferation ability of HepG2 cells induced by GSPs is achieved by regulating the cell cycle, flow cytometric analysis was conducted to assess the cell cycle distribution of HepG2 cells treated with 10 mg/L of GSPs for 24 and 48 h. The results suggested that treatment with 10 mg/L of GSPs for 24 h markedly reduced the percentage of cells in the G1 phase and elevated the percentage of cells in the S and G2 phase (Figure 4A); treatment with 10 mg/L of GSPs for 48 h significantly decreased the percentage of cells in the G1 phase and elevated the percentage of cells in G2 phase (Figure 4B). Due to the increase in the percentage of cells in the G2 and S phase, it was indicated that 10 mg/L of GSPs induced G2/M phase cell cycle arrest of the HepG2 cells. G2/M phase arrest may hinder the cell cycle and thus affect cell proliferation.

### 3.5. Effects of GSPs on Cell Cycle-Related Proteins of HepG2 Cells

Since 10 mg/L of GSPs induced G2/M phase cell cycle arrest of HepG2 cells, western blot was used to detect the expression of proteins involved in the G2/M phase. The results demonstrated that the protein expression level of cyclin B1 and CDK1 was downregulated, while p21 was upregulated for HepG2 cells treated with 10 mg/L of GSPs for 24 and 48 h (Figure 4C).

### 3.6. GSPs Induce the Phosphorylation of the MAPK Pathway-Related Proteins

The study has shown that ROS can activate MAPK pathways [43]. Thus, the expression of the MAPK pathway-related proteins was determined by western blot. The results indicated that treatment with 10 mg/L of GSPs for 24 and 48 h significantly increased the phosphorylation level of JNK, ERK and p38 MAPK of HepG2 cells (Figure 5).

### 3.7. GSPs Induce Expression of NAG-1 of HepG2 Cells

As demonstrated in Figure 6A,B, a significant increase in mRNA and protein level of NAG-1 was observed in HepG2 cells treated with 10 mg/L of GSPs for 24 and 48 h. To further investigate the relationship between GSPs-induced activation of NAG-1 and the MAPK pathway-related proteins, three inhibitors in the MAPK pathway, p38 MAPK (SB203580, 10 μM), ERK (PD98059, 20 μM) and JNK (SP600125, 10 μM) inhibitor were used to pre-incubate cells for 1 h before treatment with GSPs for 24 and 48 h. The results showed that these three inhibitors significantly attenuated the mRNA and protein expression level of NAG-1 induced by 24 and 48 h of GSPs treatment (Figure 7, Figure 8 and Figure 9). These results indicated that GSPs-induced expression of NAG-1 was associated with the MAPK pathway-related proteins.

To investigate whether NAG-1 was involved in GSPs-induced apoptosis, a NAG-1 overexpression vector was constructed to transfect HepG2 cells for 24 h and HepG2 cells were then treated with 10 mg/L of GSPs for 24 h. The results showed that transfection with the pcDNA3.1-NAG-1 plasmid for 24 h significantly increased the protein expression of NAG-1 of HepG2 cells (Figure 10A), and there was no significant difference between GSPs treatment and transfection of the pcDNA3.1-NAG-1 plasmid with GSPs treatment in early apoptosis or late apoptosis (Figure 10B,C), indicating that NAG-1 could not directly regulate GSPs-induced apoptosis.

## 4. Discussion

The imbalance between cell proliferation and cell death promotes the development of cancer [44]. Apoptosis is one of the types of programmed cell death, and it is regulated by a strict and complex signaling network [44,45]. Inducing apoptosis of tumor cells is one of the important strategies for cancer therapy [44]. In the present study, the results indicate that GSPs induced apoptosis of HepG2 cells (Figure 2), which was in accord with the previous studies that proanthocyanidins from grape seed or other plants induced apoptosis in cervical cancer [46], breast cancer [47], HCC [14], prostate cancer [35] and colorectal cancer [34].

It has been reported that changes in the level of intracellular ROS play a vital role in the early stage of apoptosis, predicting a decrease in the MMP and release of those apoptotic-inducing factors, such as cytochrome c and activated caspases [48]. Many studies have indicated that drug-induced production of ROS is the cause of apoptosis in different types of cancer [19]. Like many drugs that induce cancer cell apoptosis through ROS, studies have shown that proanthocyanidins can also induce apoptosis in cancer cells through ROS [19,20,21]. In this study, GSPs triggered ROS production, decreased the MMP, and increased the caspase-3 activity of HepG2 cells (Figure 3). Therefore, we speculate that GSPs may induce ROS production, which leads to the decrease in MMP and the activation of caspase-3, and finally leads to the apoptosis of HepG2 cells.

Cell cycle arrest is also an effective strategy for inhibiting tumor growth [49]. Previous studies demonstrated that GSPs modulated key cell cycle-related genes in colorectal cancer [50] and induced G1 phase cell cycle arrest in prostate cancer cells [35]. It has also been suggested that proanthocyanidin resulted in G2/M phase cell cycle arrest in colorectal cancer cells [34] and in HCC cells [14]. Similar to these results, we also found that 10 mg/L of GSPs significantly induced the G2/M phase cell cycle arrest of HepG2 cells (Figure 4A,B).

The G2/M phase transition is associated with the modulation of CDK-regulatory proteins and cyclin complexes [51]. p21 is pivotal in the regulation of the G2/M phase checkpoint by inhibiting the accumulation of CDK1-cyclin B complex [52]. Therefore, western blot was used to determine the expression changes of proteins associated with the G2/M phase, CDK1, cyclin B1 and p21 after GSPs treatment. Consistent with these previous research studies, cyanidin 3-glucoside and peonidin 3-glucoside resulted in G2/M phase arrest due to the downregulation of cyclin B1 and CDK1 [53,54], and lipophilic GSP, resveratrol and silybin induced cell cycle arrest by enhancing the expression of p21 in prostate cancer cells [35,55,56], we also found that GSPs treatment induced the downregulation of cyclin B1 and CDK1 and upregulation of p21 of HepG2 cells (Figure 4C).

Excess production of ROS results in increased oxidative stress, which activates intracellular signaling cascades that can lead to cellular death [57]. The study has shown that ROS can activate MAPK pathways [43]. The family of MAPK is relatively evolutionarily conserved. The MAPKs family has three major members, namely, ERK, JNK and p38 MAPK [22]. It has been found that the MAPK pathway-related proteins play an essential role in cancer growth, apoptosis and metastasis [23]. ERK is mainly involved in cell proliferation, survival, migration and invasion, and JNK and p38 MAPK are involved in cell apoptosis [58,59]. The activation of JNK and p38 MAPK is closely related to cell death [60]. Increased p38 MAPK activity can induce apoptosis of HCC cells [61]. In this work, we found that 10 mg/L of GSPs significantly increased the phosphorylation level of ERK, JNK and p38 MAPK of HepG2 cells (Figure 5), which suggests that GSPs may induce apoptosis of HepG2 via the MAPK pathway-related proteins.

NAG-1, a transforming growth factor-β superfamily member, has proapoptotic and antitumorigenic activity [24]. NAG-1 transgenic mice show anticancer effect in the colon and lung [25,26]. Dietary phytochemicals, including epicatechin gallate [30], resveratrol [62], berberine [63] and apigenin [64], enhanced the expression of NAG-1. In accordance with these studies, we also found that GSPs dramatically increased the expression of NAG-1 at the mRNA and protein level (Figure 6A,B).

It has been reported that MAPK pathway-related proteins are involved in the regulation of NAG-1 expression. The phosphorylation of p38 played an important role in stabilizing vitamin E succinate-triggered *NAG-1* mRNA expression [27]. The MAPK pathway-related proteins were involved in rottlerin-induced NAG-1 expression [28]. The GL-V9-triggered p38 MAPK pathway was conducive to the stability of *NAG-1* mRNA [29]. Similarly, our results demonstrated that GSPs-induced NAG-1 expression was associated with the MAPK pathway-related proteins, as three inhibitors in the MAPK pathway significantly attenuated the mRNA and protein expression of GSPs-induced NAG-1 (Figure 7, Figure 8 and Figure 9).

Studies have shown that NAG-1 plays a vital role in the regulation of apoptosis. Epicatechin gallate-triggered expression of NAG-1 was related to growth inhibition and apoptosis in colon cancer cells [30]. NAG-1 played an essential role in mitochondrial apoptosis induced by GL-V9 [29]. Inhibition of NAG-1 by siRNA resulted in the suppression of conjugated linoleic acid-induced apoptosis of HCT116 cells [31]. To study whether NAG-1 was involved in GSPs-induced apoptosis, a pcDNA3.1-NAG-1 overexpression vector was constructed and transfected into HepG2 cells, and cells were treated with GSPs followed by measurement of apoptosis using flow cytometry. However, inconsistent with previous results, it was found that NAG-1 could not directly regulate GSPs-induced apoptosis. Some studies indicated that NAG-1 induction by anticancer agents may involve many cellular processes other than apoptosis. Chiu et al. demonstrated that isochaihulactone-induced NAG-1 leads to cell cycle arrest, but not apoptosis, in human prostate cancer LNCaP cells [32]. Wynne et al. reported that NAG-1 induction is critical for nonsteroidal anti-inflammatory drugs-induced inhibition of cell migration in human prostate cancer PC-3 cells [33]. This study found that GSPs can induce G2/M phase cell cycle arrest of HepG2 cells (Figure 4A,B). It is speculated that GSPs-induced upregulation of NAG-1 may regulate cell growth by regulating the cell cycle and other pathways, which needs to be confirmed by further studies.

## 5. Conclusions

The present study provides evidence for the antiproliferative potential of GSPs on HepG2 cells. GSPs inhibited the viability of HepG2 cells, induced apoptosis and G2/M phase cell cycle arrest, and regulated cell cycle-related proteins. GSPs increased ROS production and caspase-3 activity. In addition, GSPs also increased the expression of the MAPK pathway-related proteins, p-ERK, p-JNK and p-p38 MAPK, and NAG-1. Furthermore, GSPs-induced NAG-1 expression was related to the MAPK pathway-related proteins. However, NAG-1 could not directly regulate GSPs-induced apoptosis. Overall, the data suggest that GSPs may be promising phytochemicals that could potentially be used to target HCC. 

## Figures and Tables

**Figure 1 antioxidants-09-01200-f001:**
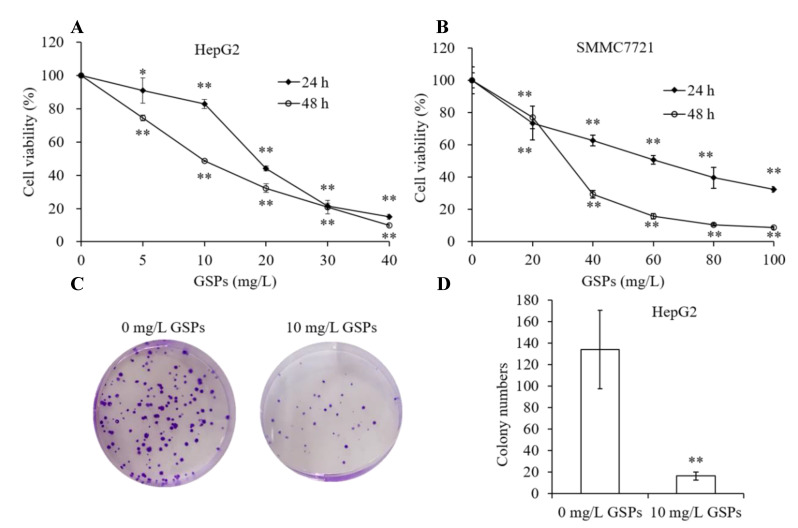
Grape seed proanthocyanidins (GSPs) inhibited the proliferation of HepG2 and SMMC-7721 cells in vitro. HepG2 (**A**) and SMMC-7721 (**B**) cells were treated with different concentrations of GSPs for 24 and 48 h, and cell viability was measured by MTT assay. (**C**,**D**) Colony formation assay for HepG2 cells treated with (10 mg/L) or without GSPs. Data are shown as the mean ± SD (*n* = 3). * and ** indicate a significant difference between the treatment group and the control group at *p* < 0.05 and *p* < 0.01, respectively.

**Figure 2 antioxidants-09-01200-f002:**
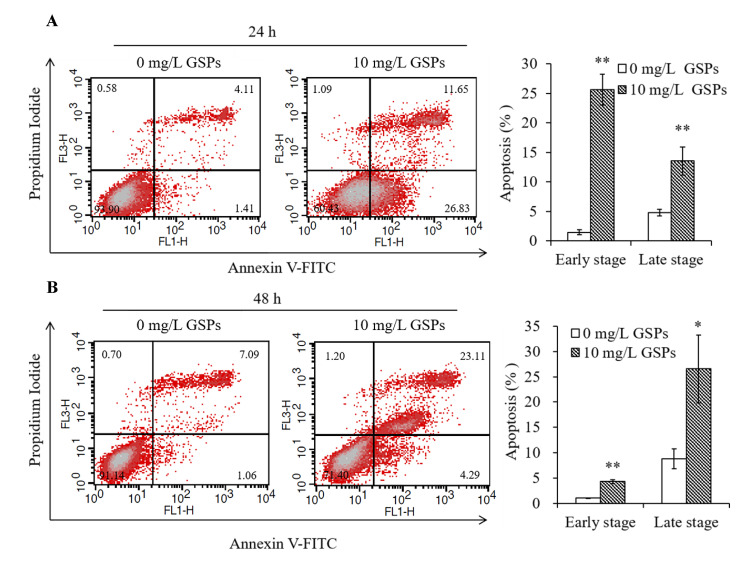
GSPs induced the apoptosis of HepG2 cells. The apoptosis of HepG2 cells treated with GSPs for 24 (**A**) and 48 h (**B**) was analyzed by flow cytometry analysis of Annexin V-FITC/PI double-stained cells. Data are shown as the mean ± SD (*n* = 3). * and ** indicate a significant difference between the treatment group and the control group at *p* < 0.05 and *p* < 0.01, respectively.

**Figure 3 antioxidants-09-01200-f003:**
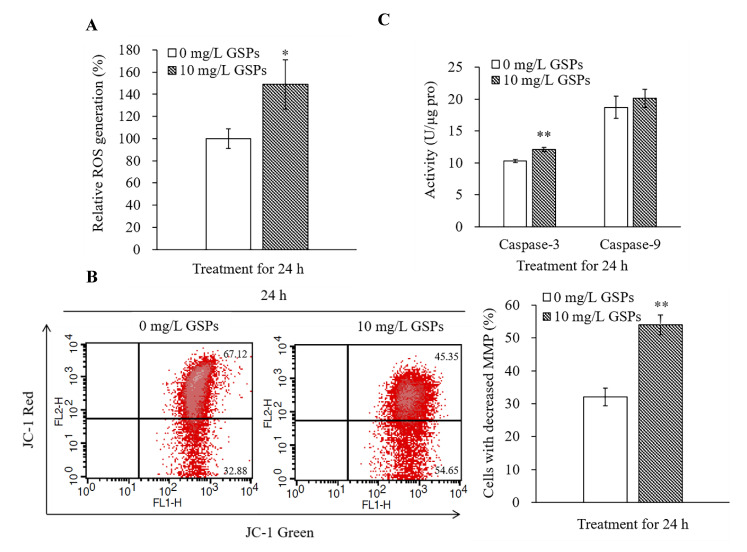
GSPs induced ROS production, induced the loss of MMP, and increased the activity of caspase-3 of HepG2 cells. HepG2 cells were treated with 10 mg/L of GSPs for 24 h, and ROS production was measured using a DCFH-DA probe (**A**). MMP was analyzed by flow cytometry with JC-1 dye (**B**), and caspase-3 and caspase-9 activity was measured using a colorimetric assay (**C**). Data are shown as the mean ± SD (*n* = 3). * and ** indicate a significant difference between the treatment group and the control group at *p* < 0.05 and *p* < 0.01, respectively.

**Figure 4 antioxidants-09-01200-f004:**
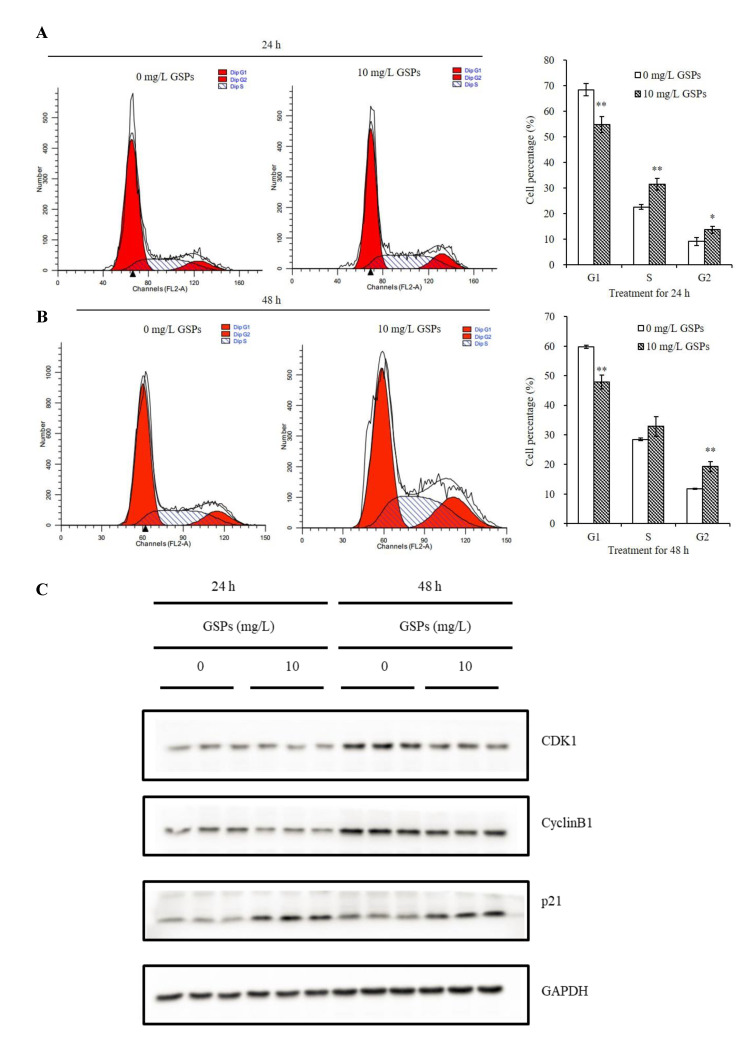
GSPs induced G2/M phase cell cycle arrest of the HepG2 cells and regulated cell cycle regulatory proteins. The cell cycle distribution of HepG2 cells treated with GSPs for 24 (**A**) and 48 h (**B**) was analyzed by flow cytometry. Data are shown as the mean ± SD (*n* = 3). Effects of GSPs on the expression of cell cycle regulatory proteins, cyclin B1, CDK1, p21 of HepG2 cells (**C**). * and ** indicate a significant difference between the treatment group and the control group at *p* < 0.05 and *p* < 0.01, respectively.

**Figure 5 antioxidants-09-01200-f005:**
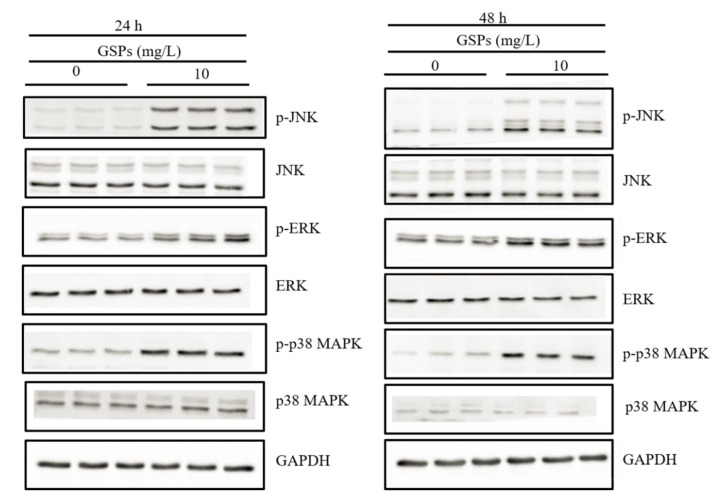
GSPs induced the activation of MAPK signaling pathways. HepG2 cells were treated with 10 mg/L of GSPs for 24 and 48 h, and the protein expression of p-JNK, JNK, p-ERK, ERK, p-p38 MAPK and p38 MAPK was detected by western blot analysis.

**Figure 6 antioxidants-09-01200-f006:**
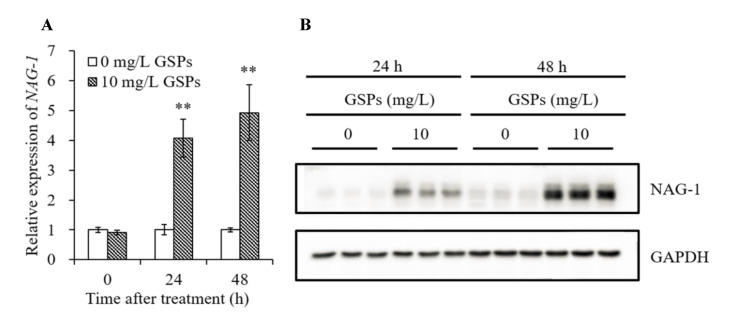
GSPs induced the expression of NAG-1 at mRNA and protein level. HepG2 cells were treated with 10 mg/L of GSPs for 24 and 48 h. The expression of NAG-1 at mRNA level was measured by qPCR analysis (**A**), and the expression of NAG-1 at protein level was detected by western blot analysis (**B**). Data are shown as the mean ± SD (*n* = 3). ** indicate a significant difference between the treatment group and the control group at *p* < 0.01.

**Figure 7 antioxidants-09-01200-f007:**
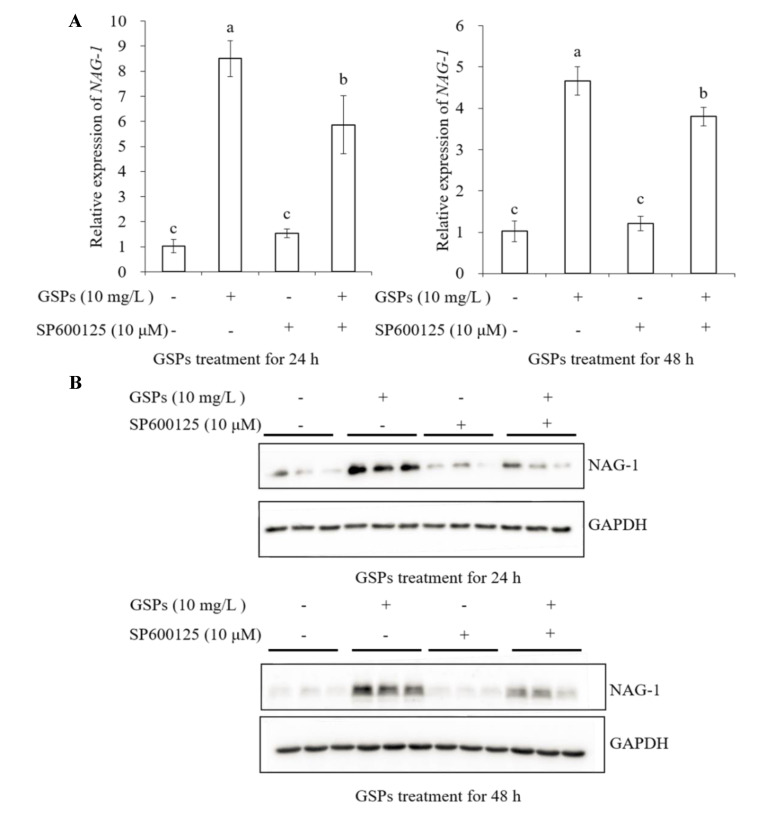
GSPs-induced NAG-1 expression was associated with the JNK signaling pathway. HepG2 cells were pre-incubated with SP600125, a JNK pathway inhibitor for 1 h, then treated with 10 mg/L of GSPs for 24 and 48 h. The expression of NAG-1 at the mRNA level was measured by qPCR analysis (**A**), and the expression of NAG-1 at the protein level was detected by western blot analysis (**B**). Data are shown as the mean ± SD (*n* = 3). Different letters indicate significant difference at *p* < 0.05.

**Figure 8 antioxidants-09-01200-f008:**
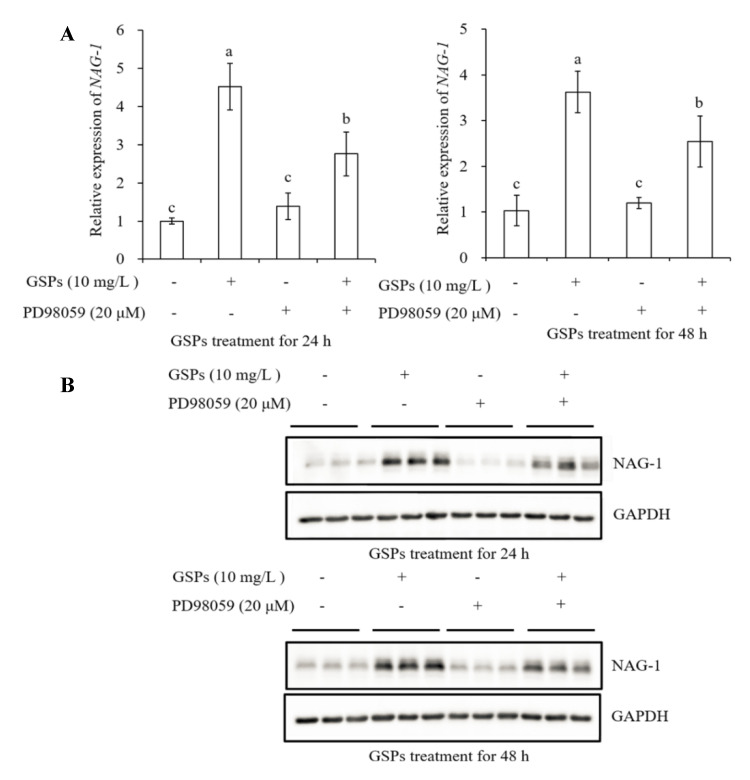
GSPs-induced NAG-1 expression was associated with the ERK signaling pathway. HepG2 cells were pre-incubated with PD98059, an ERK pathway inhibitor for 1 h, then treated with 10 mg/L of GSPs for 24 and 48 h. The expression of NAG-1 at the mRNA level was measured by qPCR analysis (**A**), and the expression of NAG-1 at the protein level was detected by western blot analysis (**B**). Data are shown as the mean ± SD (*n* = 3). Different letters indicate significant difference at *p* < 0.05.

**Figure 9 antioxidants-09-01200-f009:**
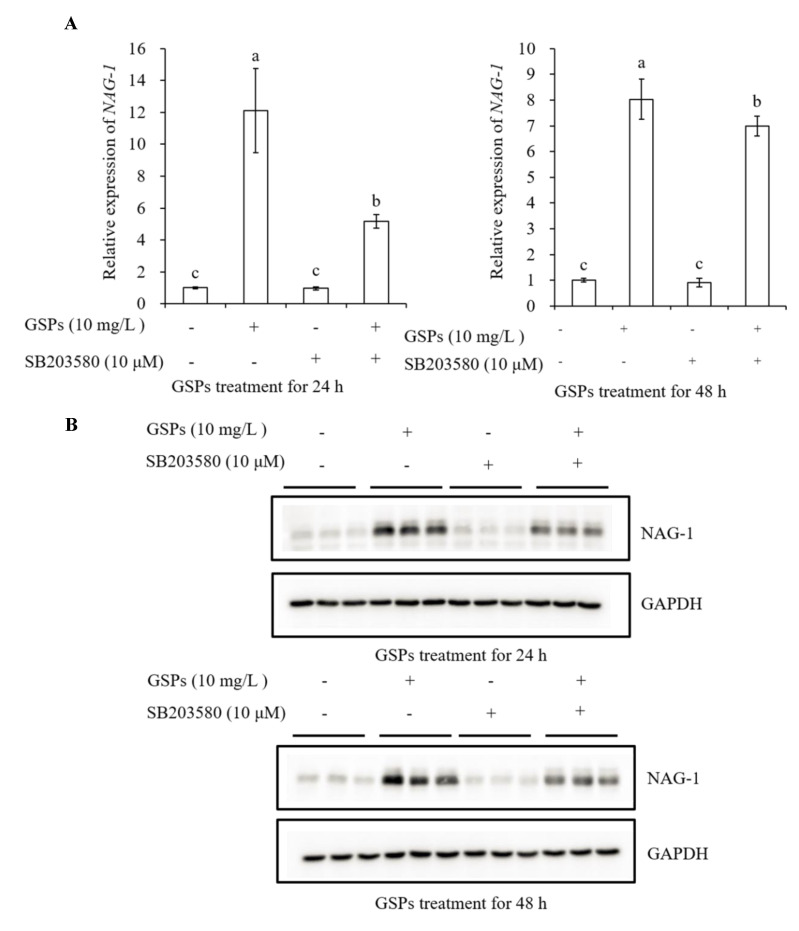
GSPs-induced NAG-1 expression was associated with the p38 MAPK signaling pathway. HepG2 cells were pre-incubated with SB203580, a p38 MAPK pathway inhibitor for 1 h, then treated with 10 mg/L of GSPs for 24 and 48 h. The expression of NAG-1 at the mRNA level was measured by qPCR analysis (**A**), and the expression of NAG-1 at the protein level was detected by western blot analysis (**B**). Data are shown as the mean ± SD (*n* = 3). Different letters indicate significant difference at *p* < 0.05.

**Figure 10 antioxidants-09-01200-f010:**
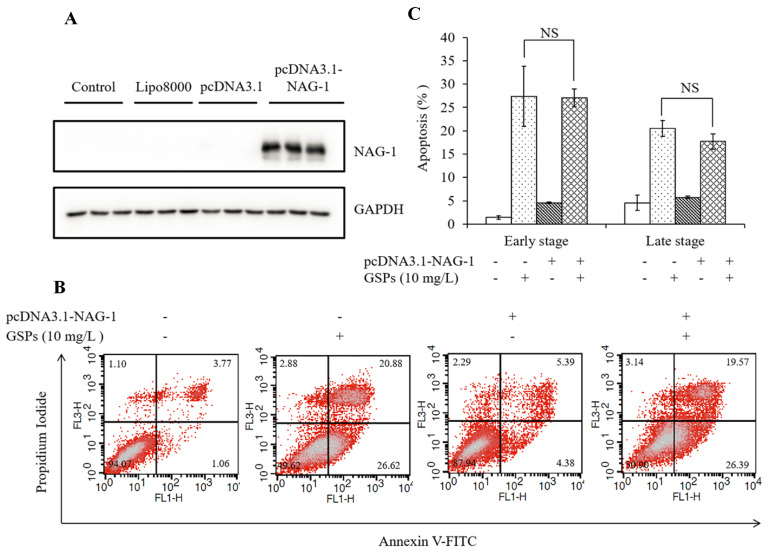
NAG-1 could not directly regulate GSPs-induced apoptosis. (**A**) HepG2 cells were transfected with pcDNA3.1-NAG-1 for 24 h, and the expression of NAG-1 at the protein level was detected by western blot analysis. (**B**) and (**C**) HepG2 cells were transfected with pcDNA3.1-NAG-1 for 24 h and then treated with 10 mg/L of GSPs for 24 h. Apoptosis of HepG2 cells was detected by flow cytometry. Data are shown as the mean ± SD (*n* = 3). NS indicates no significant difference.

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
