# Peer review of "Grape Seed Proanthocyanidins Induce Apoptosis and Cell Cycle Arrest of HepG2 Cells Accompanied by Induction of the MAPK Pathway and NAG-1"

_antioxidants, 2020, doi:10.3390/antiox9121200_

Round 1

Reviewer 1 Report

The article is good and contains relevant and high-quality results.
The methodology is well done and the results are correctly explained.

The document has some minor errors that should be corrected, as well as some doubts that have arisen during the review process:
- Lines 50-54. Authors should reference that statement
- Line 79. At what humidity and CO2 concentration were the cells incubated?
- Line 95. What were the growing conditions?
- Line 96. How were the floating cells collected and washed?
- Line 99. Please specify what the room temperature was.
- Authors should check the centigrade symbol. Sometimes they write it correctly (ºC) and sometimes they write it with the dash below. (For example, lines 106, 114, 130, 131 ...)
- Section 3.1. Why haven't you used a cell line control? (healthy) How do the authors guarantee that these compounds attack cancer cells and not normal cells?

Author Response

Response to the reviewer 1 comments

Comments and Suggestions for Authors

The article is good and contains relevant and high-quality results.

The methodology is well done and the results are correctly explained.

The document has some minor errors that should be corrected, as well as some doubts that have arisen during the review process:

- Lines 50-54. Authors should reference that statement

Answer: Thank you for your helpful suggestion. We have added reference in the revised manuscript (lines 58-59).

- Line 79. At what humidity and CO2 concentration were the cells incubated?

Answer: Thank you for your helpful suggestion. We have added humidity and CO2 concentration for cell incubation in the revised manuscript (line 88).

- Line 95. What were the growing conditions?

Answer: Thank you for your helpful suggestion. We have added growing conditions for cells in the revised manuscript (lines 104-105).

- Line 96. How were the floating cells collected and washed?

Answer: Thank you for your helpful suggestion. We have added the method of how to collect and wash floating cells in the revised manuscript (lines 106-107).

- Line 99. Please specify what the room temperature was.

Answer: Thank you for your helpful suggestion. We have specified what the room temperature used in the revised manuscript (line 109).

- Authors should check the centigrade symbol. Sometimes they write it correctly (ºC) and sometimes they write it with the dash below. (For example, lines 106, 114, 130, 131 ...)

Answer: Thank you for your helpful suggestion. We have carefully checked all the centigrade symbol in the revised manuscript.

- Section 3.1. Why haven't you used a cell line control? (healthy) How do the authors guarantee that these compounds attack cancer cells and not normal cells?

Answer: In fact, we have previously investigated the effects of 10 mg/L and 20 mg/L of GSPs on the viability of L02 human normal liver cells, and the results are shown in Figure 1 below. The results showed that treatment of 10 mg/L and 20 mg/L of GSPs for 24 h or 48 h basically did not reduce the cell viability of L02.

Figure 1 Effects of different concentrations of GSPs on the cell viability of L02 human normal liver cells

In addition, whether the dose of GSPs (100 mg/kg and 200 mg/kg) displayed observable toxicity in nude mice was also evaluated in our previous study (Wang et al., 2019). Changes in the body weight of the nude mice during gavage with GSPs; two indicators of liver function, namely alanine aminotransferase (ALT) and aspartate aminotransferase (AST); an indicator of kidney function, creatinine (Cr); and HE staining of the liver were evaluated. The results showed that GSPs doses at 100 mg/kg and 200 mg/kg did not significantly affect body weight, ALT, AST, and Cr in nude mice, while the HE staining showed no damage to the liver of nude mice, indicating that the doses of GSPs at 100 mg/kg and 200 mg/kg caused no observable toxicity in nude mice.

Wang L.H., Huang W.D., Zhan J.C. Grape seed proanthocyanidins induce autophagy and modulate survivin in HepG2 cells and inhibit xenograft tumor growth in vivo. Nutrients, 2019, 11(12), 2983.

Reviewer 2 Report

Overall, this is a well-presented and organized study. The results support the conclusions reached by the authors. There a few minor points that the authors need to address. The points are listed below in order of appearance in the manuscript.

  1. page 2, line 59: There are two green highlighted portions of text and need to be deleted.
  2. page 3, lines 103-109: Was a calibration curve generated to make the quantitative measurements? If so the standards/methodology needs to be included here.
  3. page 4, lines 140-142: Again, there is green highlighted portions of text that need to be deleted.
  4. page 5, figure 1: I suggest changes the symbols in figure 1a and 1b. Maybe used open circles and filled diamonds, as the symbols are a bit difficult to see.
  5. page 16, line 449: There is green highlighted portions of text that need to be deleted.
  6. page 16, line 465: reference number 19 is in red and should be black and there is green highlighted text.
  7. page 17, line 495: reference number 30 is in red and should be black 
  8. page 17, line 518: reference number 38 is in red and should be black 
  9. page 18, lines 546, 567: reference number 47 is in red and should be black and there is green highlighted text 

Author Response

Response to the reviewer 2 comments

Comments and Suggestions for Authors

Overall, this is a well-presented and organized study. The results support the conclusions reached by the authors. There a few minor points that the authors need to address. The points are listed below in order of appearance in the manuscript.

page 2, line 59: There are two green highlighted portions of text and need to be deleted.

Answer: We have carefully checked the two green highlighted portions of text in line 59 of original manuscript (Dulbecco's Modifed Eagle's Medium). And two green highlighted portions of text should not be deleted.

We have searched some references, DMEM can be written as Dulbecco's Modified Eagle's Medium (Lim et al., 2012; Wu et al., 2018; Yang et al., 2017).

Lim, J.H.; Woo, S.M.; Min, K.J.; Park, E.J.; Jang, J.H.; Seo, B.R.; Iqbal, T.; Lee, T.J.; Kim, S.H.; Choi, Y.H., et al. Rottlerin induces apoptosis of HT29 colon carcinoma cells through NAG-1 upregulation via an ERK and p38 MAPK-dependent and PKC δ-independent mechanism. Chemico-Biological Interactions 2012, 197, 1-7, doi:10.1016/j.cbi.2012.02.003.

Wu, K.; Na, K.; Chen, D.; Wang, Y.; Pan, H.; Wang, X. Effects of non-steroidal anti-inflammatory drug-activated gene-1 on Ganoderma lucidum polysaccharides-induced apoptosis of human prostate cancer PC-3 cells. International Journal of Oncology 2018, 53, 2356-2368, doi:10.3892/ijo.2018.4578.

Yang, N.; Gao, J.; Cheng, X.; Hou, C.; Yang, Y.; Qiu, Y.; Xu, M.; Zhang, Y.; Huang, S. Grape seed proanthocyanidins inhibit the proliferation, migration and invasion of tongue squamous cell carcinoma cells through suppressing the protein kinase B/nuclear factor-κ B signaling pathway. International Journal of Molecular Medicine 2017, 40, 1881-1888, doi:10.3892/ijmm.2017.3162.

page 3, lines 103-109: Was a calibration curve generated to make the quantitative measurements? If so the standards/methodology needs to be included here.

Answer: A calibration curve is not used here to quantitatively measure ROS production. ROS production is calculated in a relatively quantitative way, that is, the control group is defined as 100% (Figure 3A).

page 4, lines 140-142: Again, there is green highlighted portions of text that need to be deleted.

Answer: We have carefully checked the green highlighted portions of text in line 140-142 of original manuscript (NAG-1-sense (5'-CAGTCGGACCAACTGCTGGCA-3'), NAG-1-antisense (5'-TGAGCACCATGGGATTGTAGC-3'); GAPDH-sense (5'-TCTGGTAAAGTGGATATTGTTG-3'), GAPDH -antisense (5'-GATGGTGATGGGATTTCC-3')). And these green highlighted portions of text should not be deleted.

We have searched some reference, primer sequence can be written as 5'-CAGTCGGACCAACTGCTGGCA-3' (Baek et al., 2002; Zhang et al., 2017).

Zhang, X.B.; Kang, Y.; Huo, T.X.; Tao, R.; Wang, X.P.; Li, Z.Y.; Guo, Q.L.; Zhao, L. GL-V9 induced upregulation and mitochondrial localization of NAG-1 associates with ROS generation and cell death in hepatocellular carcinoma cells. Free Radical Biol. Med. 2017, 112, 49-59. doi:10.1016/j.freeradbiomed.2017.07.011.

Baek, S.J.; Wilson, L.C.; Eling, T.E. Resveratrol enhances the expression of non-steroidal anti-inflammatory drug-activated gene (NAG-1) by increasing the expression of p53. Carcinogenesis 2002, 23, 425-434. doi:10.1093/carcin/23.3.425.

page 5, figure 1: I suggest changes the symbols in figure 1a and 1b. Maybe used open circles and filled diamonds, as the symbols are a bit difficult to see.

Answer: Thank you for your helpful suggestion. We have changed the symbols in figure 1a and 1b to open circles and filled diamonds to better show the figure.

page 16, line 449: There is green highlighted portions of text that need to be deleted.

Answer: We have carefully checked the green highlighted portions of text in line 449 of original manuscript (Molecular characterization of the grape seeds extract's effect against chemically induced liver cancer: In vivo and in vitro analyses). And the green highlighted portions of text should not be deleted.

The title of this reference is ‘Molecular characterization of the grape seeds extract's effect against chemically induced liver cancer: In vivo and in vitro analyses’, as shown in the link below.

https://apps.webofknowledge.com/Search.do?product=UA&SID=5ChnKXCGVHF9wDCChde&search_mode=GeneralSearch&prID=1d71c854-ed4e-47ba-b6f6-2e60d42f2f13

page 16, line 465: reference number 19 is in red and should be black and there is green highlighted text.

page 17, line 495: reference number 30 is in red and should be black

page 17, line 518: reference number 38 is in red and should be black

page 18, lines 546, 567: reference number 47 is in red and should be black and there is green highlighted text

Answer: Thank you for your helpful suggestion. We have carefully checked these similar font color and changed them to black color.

Reviewer 3 Report

This manuscript described the effect of grape seed proanthocyanidins on the apoptosis, cell cycle and MAPK andNAG-1 expressions of HepG2 cells. The study is well designed and interesting. However, there are some questions and comments as below:

1.  P2 line 62, the raw material GSPs used in this study are multi-components, not a single component. The composition was not described in the text. The author should describe its component composition or perform a component analysis to find out its main component.

2.  The western blot image in Figure 4 should be quantified as in Figure 6 and Figure 7 to make the result clearer.

3. Figure 5

(1) Figure 5 should be quantified to make the results clearer.

(2) Both p-JNK and JNK graphs have two bands, which one is correct?

Author Response

Response to the reviewer 3 comments

Comments and Suggestions for Authors

This manuscript described the effect of grape seed proanthocyanidins on the apoptosis, cell cycle and MAPK and NAG-1 expressions of HepG2 cells. The study is well designed and interesting. However, there are some questions and comments as below:

  1. P2 line 62, the raw material GSPs used in this study are multi-components, not a single component. The composition was not described in the text. The author should describe its component composition or perform a component analysis to find out its main component.

Answer: Thank you for your helpful suggestion. It has been reported that GSPs are mixture of dimers, trimers, tetramers, and oligomers/polymers by the polymerization of catechins and/or epicatechins (Lan et al., 2015). We described the composition of GSPs briefly in the section of introduction (lines 41-43).

Lan, C.Z.; Ding, L.; Su, Y.L.; Guo, K.; Wang, L.; Kan, H.W.; Ou, Y.R.; Gao, S. Grape seed proanthocyanidins prevent DOCA-salt hypertension-induced renal injury and its mechanisms in rats. Food Funct. 2015, 6, 2179-2186. doi:10.1039/c5fo00253b.

  1. The western blot image in Figure 4 should be quantified as in Figure 6 and Figure 7 to make the result clearer.

Answer: The bar graphs in Figure 6 and Figure 7 are the expression of mRNA levels, not the quantitative results of western blot. The differences of between treatment and control are significant from visual observation in western blot image, and western blot results show 3 replicate gel wells. So, we did not quantify the western blot band intensities.

  1. Figure 5

(1) Figure 5 should be quantified to make the results clearer.

Answer: The differences of between treatment and control are significant from visual observation in western blot image, and western blot results show 3 replicate gel wells. So, we did not quantify the western blot band intensities.

(2) Both p-JNK and JNK graphs have two bands, which one is correct?

Answer: Both p-JNK and JNK contain two bands in western blot image, as shown in the antibody product description below.

p-JNK

https://www.cellsignal.cn/products/primary-antibodies/phospho-sapk-jnk-thr183-tyr185-81e11-rabbit-mab/4668?_=1540972685425&Ntt=4668t&tahead=true

JNK

https://www.ptgcn.com/Products/JNK-Antibody-24164-1-AP.htm

Reviewer 4 Report

The study “Grape seed proanthocyanidins induce apoptosis and cell cycle arrest of HepG2 cells accompanied by induction of the MAPK pathway and NAG-1” by Wang and co-workers investigates the effect of GSPs in HepG2 cells in the context of the hepatocellular carcinoma.

In the introduction section essential information is missing. Authors should include information on NAG-1 and MAPK.

No information on incubation period of cells with GSPs for ROS detection, MMP, capsase activity, real time PCR, western blot is availabel. However, at e.g.figure 6 it is mentioned that mRNA levels have been measured after 24 h and 48 h of treatment. This is also true for the protein expression detected with western blot. Does this make sense? To detect protein expression after the same incubation period than mRNA levels? The authors should comment on that.

Missing primer information on GAPDH.

Why have the authors sticked to 10 mg/l GSP treatment of HepG2 cells? Why have they not included other concentrations? The cytotoxicity assay was performed for both cell lines (HepG2 and SMCC-7721). In the following experiments the authors only looked into effects on HepG2 cells. Why have authors focused only on HepG2? In case there is no reason for focusing on HepG2 cells SMCC-7721 can be excluded from the manuscript to make it more clearly. In case there is a reason it should be mentioned/discussed.

The discussion section starts with mentioning the different GSP concentrations that have been applied within other cell studies. This information is not necessary in deep detail and should be removed/revised.

The information on the performance of Western blotting is incomplete.

Figures need to be increased in size/fonts.

Author Response

Response to the reviewer 4 comments

Comments and Suggestions for Authors

The study “Grape seed proanthocyanidins induce apoptosis and cell cycle arrest of HepG2 cells accompanied by induction of the MAPK pathway and NAG-1” by Wang and co-workers investigates the effect of GSPs in HepG2 cells in the context of the hepatocellular carcinoma.

In the introduction section essential information is missing. Authors should include information on NAG-1 and MAPK.

Answer: Thank you for your helpful suggestion. We have added information on NAG-1 and MAPK in the introduction section in the revised manuscript (lines 50-57).

No information on incubation period of cells with GSPs for ROS detection, MMP, capsase activity, real time PCR, western blot is available. However, at e.g.figure 6 it is mentioned that mRNA levels have been measured after 24 h and 48 h of treatment. This is also true for the protein expression detected with western blot. Does this make sense? To detect protein expression after the same incubation period than mRNA levels? The authors should comment on that.

Answer: We have added information on incubation period of cells with GSPs for ROS detection, MMP, capsase activity, real time PCR and western blot in the section of methods in the revised manuscript.

Due to the existence of post-transcriptional regulation, sometimes there are inconsistencies between the expression of mRNA level and protein level. The simultaneous determination of the expression of mRNA level and protein level is mainly to judge which level of gene regulation is regulated, which provides a basis for further study on molecular mechanisms at the transcription level and protein level.

Missing primer information on GAPDH.

Answer: We have added the primer information on GAPDH in section 2.9 in the revised manuscript (lines 153-154).

Why have the authors sticked to 10 mg/l GSP treatment of HepG2 cells? Why have they not included other concentrations? The cytotoxicity assay was performed for both cell lines (HepG2 and SMCC-7721). In the following experiments the authors only looked into effects on HepG2 cells. Why have authors focused only on HepG2? In case there is no reason for focusing on HepG2 cells SMCC-7721 can be excluded from the manuscript to make it more clearly. In case there is a reason it should be mentioned/discussed.

Answer: In section 3.1, it is explained why the 10 mg/L GSPs treatment of HepG2 cells was selected. and why the HepG2 cells are further studied.

As shown in Figure 1a and b, GSPs reduced the viability of HepG2 and SMMC-7721 cells in a dose- and time-dependent manner. Since treatment of GSPs showed more significant inhibition on HepG2 cells, the HepG2 cells were selected as the cell line for further study. When HepG2 cells were treated with 10 mg/L of GSPs for 48 h, cell viability was reduced by approximately 50%, so 10 mg/L of GSPs was selected as the treatment concentration for further study.

The discussion section starts with mentioning the different GSP concentrations that have been applied within other cell studies. This information is not necessary in deep detail and should be removed/revised.

Answer: Thank you for your helpful suggestion. We have removed this section in revised manuscript.

The information on the performance of Western blotting is incomplete.

Answer: Thank you for your helpful suggestion, we have completed information on the performance of western blotting.

Figures need to be increased in size/fonts.

Answer: Thank you for your helpful suggestion. We have adjusted the size/fonts of the figures.

Round 2

Reviewer 4 Report

The authors have addressed most of my comments. However, there was only a change in figure 3 regarding size and fonts. Especially figure 7 - which now looks even smaller than in the first version of the manuscript - is hard to read and should be revised.

Author Response

Response to the reviewer 4 comments

Comments and Suggestions for Authors

The authors have addressed most of my comments. However, there was only a change in figure 3 regarding size and fonts. Especially figure 7 - which now looks even smaller than in the first version of the manuscript - is hard to read and should be revised.

Answer: Thank you for your helpful suggestion. We have adjusted size and fonts of all figures (Figure 1- Figure 8) in the revised manuscript.

This manuscript is a resubmission of an earlier submission. The following is a list of the peer review reports and author responses from that submission.